# Effect of Fe Content on the As-Cast Microstructures of Ti–6Al–4V–xFe Alloys

**Ling Ding [1], Rui Hu [2], Yulei Gu [1], Danying Zhou [1], Fuwen Chen [1], Lian Zhou [1] and Hui Chang [1,\*]**

[1] Tech Institute for Advanced Materials & College of Materials Science and Engineering, Nanjing Tech University, Nanjing 210009, China; dingling2013@njtech.edu.cn (L.D.); 825721910@njtech.edu.cn (Y.G.); zhoudanying@njtech.edu.cn (D.Z.); fuwenchen@njtech.edu.cn (F.C.); zhoul@c-nin.com (L.Z.)

[2] State Key Laboratory of Solidification Processing, Northwestern Polytechnical University, Xi'an 710072, China; rhu@nwpu.edu.cn

\* Correspondence: ch2006@njtech.edu.cn; Tel.: +86-13813916521

**Abstract:** In this work, the evolution of the solidification microstructures of Ti–6Al–4V–xFe (x = 0.1, 0.3, 0.5, 0.7, 0.9) alloys fabricated by levitation melting was studied by combined simulative and experimental methods. The growth of grains as well as the composition distribution mechanisms during the solidification process of the alloy are discussed. The segregation of the Fe element at the grain boundaries promotes the formation of a local composition supercooling zone, thus inhibiting the mobility of the solid–liquid interface and making it easier for the grains to grow into dendrites. With the increase in Fe content, the grain size of the alloy decreased gradually, while the overall decreasing trend was mitigated. The segregation of Fe was more obvious than that of Al and V, and the increase in Fe content had less effect on the segregation of Al and V.

**Keywords:** titanium alloy; simulation; boundary; segregation

## 1. Introduction

Titanium alloys have become excellent structural materials in many fields in recent years, especially in the field of aerospace applications due to their high specific strength, corrosion resistance, and other advantages [1]. Fe, as a common β-eutectoid alloy element in titanium alloys, which is even stronger than Cr, has a great influence on the solid/liquid transformation point. The increase in Fe content may cause a β-spot. Generally, the Fe content in titanium alloy is less than 5.5 wt% [2].

According to the previous research, the mechanical properties of titanium alloys can be effectively improved by adding an appropriate amount of Fe [3,4]. Kudo et al. studied the influence of microstructure on the formability of a Ti–Fe alloy [5] and found that the formability of a Ti–Fe alloy increased with the decrease in the size of the prior β phase region. Bermingham et al. found that the addition of an appropriate amount of Fe can effectively refine the grains of titanium alloys [6]. It was considered that the segregation of Fe provided the undercooling needed to inhibit grain growth and activate adjacent nuclei. It is obvious that the addition of an appropriate amount of Fe in the titanium alloys can affect the morphology of the original beta grains during solidification, thus influencing the mechanical properties. Ehtemam designed and manufactured Ti–11Nb–xFe (x = 0.5, 3.5, 6, 9 wt%) alloys by cold crucible levitation melting to study the effect of Fe addition on its phase transformation, microstructure, and mechanical properties [7]. The results showed that the Ti–11Nb–0.5Fe alloy had a typical dual phase microstructure of α + β and the volume fraction of the β phase could be increased by increasing the Fe content. However, the formation and growth of the original beta grains during the solidification process of titanium alloys are difficult to observe experimentally, so it is not easy to verify the mechanism of Fe on the grain morphology.

Through the phase field simulation, the microstructure evolution during the solidification process as well as the influence of element content on the microstructure of the alloys can be examined. The phase field model is a powerful tool to describe the complex evolution of the interface between the matrix and new phases in the non-equilibrium state based on the unified control equations in the whole system [8,9], which is suitable for describing solid–liquid phase transformation [10]. However, the simulation of microstructure evolution with the phase field method relies on the data of the temperature field parameters and thermophysical parameters of the related elements. The electromagnetic-thermal coupled simulation conducted by Kermanpur et al. [11] and the multi physical field coupling simulation conducted by Li [12] provided the data needed for the temperature field of the microstructure simulation.

For the simulation of microstructure, Kundin et al. used the phase field method to simulate the solidification of the Ti–Fe alloy [13], and Gong et al. studied the microstructure evolution of a Ti–6Al–4V alloy by the phase field method [14–16]. As for the related thermophysical parameters, Nakajima used the tracer diffusion method and Mossbauer spectrum to study the diffusion of Fe in the β-titanium alloy [17]. It is considered that the diffusion mechanism of Fe in β-titanium alloy is an extremely rapid interstitial diffusion. Chen et al. used the DICTRA software (Thermo-Calc Software Solna, Sweden) to strictly evaluate the experimental diffusion data to determine the atomic mobility of the BCC phase in the Ti–Al–Fe system [18]. Through the comprehensive comparison between the calculated and the experimental diffusion coefficients, a better consistency is obtained. The developed mobility of atoms is verified by good prediction of the mutual diffusion behavior observed in the diffusion couple experiment in the existing literatures.

In this work, the effect of Fe content on the microstructure of Ti–6Al–4V–xFe (x = 0.1, 0.3, 0.5, 0.7, 0.9) alloys produced by levitation melting was studied by the phase field method and verified by experiments. Levitation melting is often used in the laboratory research of titanium alloys due to the small size and uniform composition of the ingot. The Ti–6Al–4V alloy is the most widely used titanium alloy (α + β type) with good comprehensive mechanical properties, which is composed of a vanadium rich BCC phase (body centered cubic, β) and aluminum rich HCP phase (hexagonal close packed, α) [19].

## 2. Model and Experiments

### 2.1. Phase Field Model

As the temperature change calculated according to the simulation is small, the following assumptions were made:

(1)   The diffusion coefficients of Al, V, and Fe in the solid phase and the liquid phase did not change in the simulation.
(2)   The temperature gradient and cooling rate in the whole process remained invariant.

Dendritic growth and grain growth models were established using MICRESS 6.3 (ACCESS e.V. Aachen, Germany) software. The dendrite growth model had a mesh size of $600 \times 600$, a cell resolution of 0.1 μm, and a minimum time step of $1 \times 10^{-3}$ s. The initial condition was considered to be 1 for the initial grain. The grain growth model had a grid size of $1000 \times 1000$, a cell resolution of 5 μm, and a minimum time step of $1 \times 10^{-2}$ s. Figure 1 shows a schematic of the modeled domain, which was in the middle of ingot. Set 10 initial grain levels to randomly generate grains according to grain radius and distribution density.

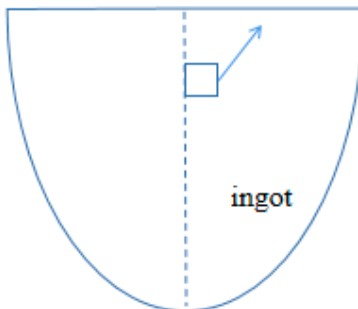

**Figure 1.** Schematic of the modeled domain.

Multiphase field theory is a computational method to describe the evolution of multiphase field parameters $\varphi_\alpha(\vec{x}, t)$ in time and space [20]. At the solid–liquid interface, 0 and 1 represent the liquid phase and solid phase, respectively, and $\varphi_\alpha$ changes continuously between 0 and 1 with an interface thickness $\eta$. Based on the principle of minimum free energy, the multiphase field equation of MICRESS was used [21]:

$$\dot{\varphi}_\alpha = \sum_\beta M_{\alpha\beta}(\vec{n})\left(\sigma^*_{\alpha\beta}(\vec{n})K_{\alpha\beta} + \frac{\pi}{\eta}\sqrt{\varphi_\alpha\varphi_\beta}\Delta G_{\alpha\beta}(\vec{c}, T)\right) \tag{1}$$

$$K_{\alpha\beta} = \varphi_\beta\nabla^2\varphi_\alpha - \varphi_\alpha\nabla^2\varphi_\beta + \frac{\pi^2}{\eta^2}(\varphi_\alpha - \varphi_\beta) \tag{2}$$

where $M_{\alpha\beta}$ is the mobility of the interface of the interface orientation, described by the normal vector $\vec{n}$. $\sigma^*_{\alpha\beta}$ is the effective anisotropic surface energy, and $K_{\alpha\beta}$ is about the local curvature of the interface. $\Delta G_{\alpha\beta}$ is the thermodynamic driving force, which is a function of the composition $\vec{c}$, and the diffusion equation can be described as:

$$\dot{\vec{c}} = \nabla\sum_{\alpha=1}^{N}\varphi_\alpha\vec{D}_\alpha\nabla\vec{c}_\alpha \tag{3}$$

where $\vec{D}_\alpha$ is the multicomponent diffusion coefficient matrix for phase $\alpha$.

The boundary conditions are based on the symmetric boundary of the MICRESS software. The phase field value of the boundary element is defined to be the same as the second adjacent element in the analog domain, thereby revealing that a plane of symmetry is crossing through the center of the outermost element of the region. This condition is similar to an isolation condition that moves half a unit. The interface thickness is 5 cells.

The simulated interface energy can use common interface energy [22]. The phase diagram data required for the simulation are directly extracted from the Thermo-Calc 2015b (Thermo-Calc Software Solna, Sweden) TTTi3 database. The solid phase diffusion coefficients of the Al and V are calculated from the MOBTI1 database, and the liquid phase diffusion coefficients of Al and V are estimated. Since there are no diffusion data of Fe in the MOBTI1 database, a kinetic database containing Fe was prepared by Chen's study of β phase diffusion kinetics of a Ti–Al–Fe alloy [18], and the data obtained were imported into MICRESS to calculate the solid phase diffusion coefficient of Fe. The liquid phase diffusion coefficient of Fe was derived from the solid phase diffusion coefficient of Fe with reference to Kundin's study [13]. The partitial physical parameters are shown in Table 1.

**Table 1.** Partial physical parameters [13,18,22].

| Physical Parameters | Ti–6Al–4V–xFe |
| --- | --- |
| Interface energy $\sigma$ (J/cm$^2$) | $2 \times 10^{-5}$ |
| Al Liquid diffusion coefficient $D_l$ (cm$^2$/s) | $1.5 \times 10^{-5}$ |
| Al Solid diffusion coefficient $D_s$ (cm$^2$/s) | $1.3 \times 10^{-7}$ |
| V Liquid diffusion coefficient $D_l$ (cm$^2$/s) | $5 \times 10^{-5}$ |
| V Solid diffusion coefficient $D_s$ (cm$^2$/s) | $6.9 \times 10^{-7}$ |
| Fe Liquid diffusion coefficient $D_l$ (cm$^2$/s) | $1 \times 10^{-4}$ |
| Fe Solid diffusion coefficient $D_s$ (cm$^2$/s) | $2 \times 10^{-5}$ |
| Molar volume $V$ (cm$^3$/mol) | 11.2 |
| Calculated temperature $T$ (K) | 1950 |
| Anisotropic strength $\eta$ | 0.05 |

## 2.2. Experiments

The chemical composition of the Ti–6Al–4V–xFe samples (melted by Levitation melting to obtain a hemispherical ingot of about 800 g with a diameter of 90 mm, furnace cooling) is shown in Table 2.

**Table 2.** Mass fraction of each element.

| Alloys | Al (wt%) | V (wt%) | Fe (wt%) | O (wt%) |
| --- | --- | --- | --- | --- |
| Ti–6Al–4V | 5.98 | 4.10 | 0.03 | 0.083 |
| Ti–6Al–4V–0.1Fe | 5.92 | 4.05 | 0.13 | 0.110 |
| Ti–6Al–4V–0.3Fe | 5.99 | 4.09 | 0.33 | 0.084 |
| Ti–6Al–4V–0.5Fe | 5.95 | 4.07 | 0.52 | 0.076 |
| Ti–6Al–4V–0.7Fe | 5.92 | 4.02 | 0.73 | 0.081 |
| Ti–6Al–4V–0.9Fe | 5.99 | 4.10 | 0.91 | 0.033 |

A 5 mm thick flat plate was cut by wire electrode cutting in the middle of the ingot. Six $15 \times 15$ mm squares were cut from the center of the ingots. The samples were electrolytic polished (using HClO$_4$:C$_2$H$_5$OH = 3:57 electrolyte) and quickly washed in alcohol and distilled water.

The metallographic photographs were obtained with an optical microscope (OM, Carl Zeiss, Jena, Germany). Line scan and surface scan images of the grain boundary of the Ti–6Al–4V–xFe alloys were obtained by electron probe micro analysis (EPMA, JEOL, Tokyo, Japan). As the primary β grain of the alloy is larger and the grain boundary is finer, the grain boundary is easily confused with the precipitated α lamellae structure, making it difficult to find the grain boundary in backscattered electron (BSE) mode. However, electropolishing (electropolishing is slightly corrosive) and secondary electron image (SEI) mode are used to find the original β grain boundary. Due to the precision limitation of EPMA, when the Fe content is low, it is difficult to measure it accurately. Therefore, Ti–6Al–4V–0.5Fe and Ti–6Al–4V–0.9Fe alloys were selected for surface scanning on the triangular crystal surface, and Ti–6Al–4V–0.5Fe, Ti–6Al–4V–0.7Fe, and Ti-6Al-4V-0.9Fe alloys were selected for line scanning through the grain boundary to obtain the corresponding element concentration distribution.

## 3. Results and Discussions

### 3.1. Effect of Fe Content on the Microstructure of Single Crystal

First, we studied the growth of the single grain. In the process of the alloy growing, the solute concentration in the liquid phase at the front of the solid–liquid interface decreased with the increase in distance from the interface, and the corresponding liquidus temperature $T_L$ changed from low to high. When the curve of the liquidus temperature $T_L$ was higher than the actual temperature $T_Q$ line in the liquid phase, the composition supercooled zone will be formed in the liquid phase at the front of the solid–liquid interface.

With the solidification layer moving inward, the heat dissipation ability of the solid phase was gradually weakened. The internal temperature gradient tended to be gentle. The solute atoms in the liquid phase were enriched, so the component supercooling in front of the interface increased. As the distribution coefficient of Al and V elements is close to that of Ti and their content is relatively low, the alloy is similar to pure metal if there is no Fe element in the alloy. Therefore, the component supercooling was not obvious and the grain was nearly plane growth, as shown in Figure 2a (the color bar represents the field parameters field parameters $\varphi$, and 0 and 1 represent the liquid phase and solid phase respectively). When the Fe content increased to 0.9 wt%, the component supercooled region at the front of the interface was larger. The protruding part continued to grow into the supercooled liquid phase. At the same time, branches grew on its side, and the grain growth tended to be dendrite. With the increase in Fe content, the growth rate of the whole grain decreased. In a certain concentration range, Fe content has a great influence on the morphology of Ti–6Al–4V grains. As shown in Figure 3, in the early stage of solidification, the grain surface was relatively stable. The solid surface formed a bulge and gradually extended with time to the supercooled zone. Due to the small temperature gradient (5 K/cm) of the suspension melting, equiaxed grains were finally formed.

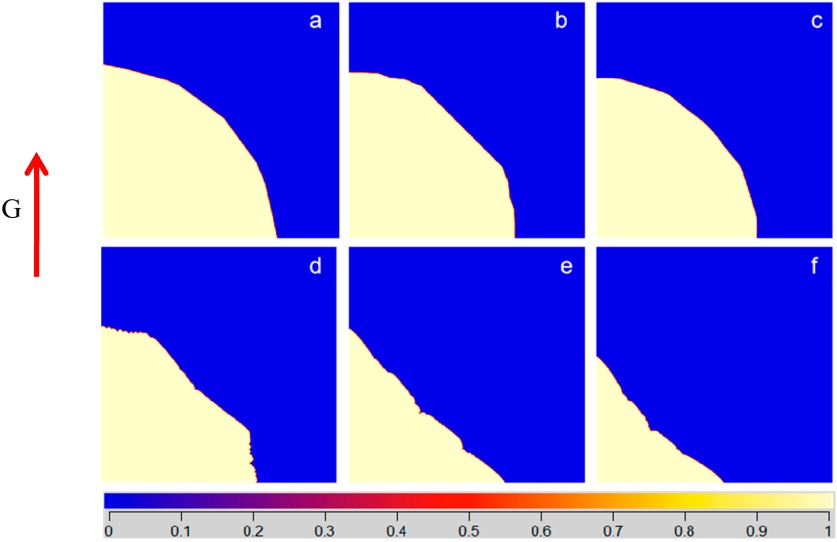

**Figure 2.** Effect of Fe content on the morphology of a single grain: (**a**) Ti–6Al–4V, (**b**) Ti–6Al–4V–0.1Fe, (**c**) Ti–6Al–4V–0.3Fe, (**d**) Ti–6Al–4V–0.5Fe, (**e**) Ti–6Al–4V–0.7Fe, (**f**) Ti–6Al–4V–0.9Fe.

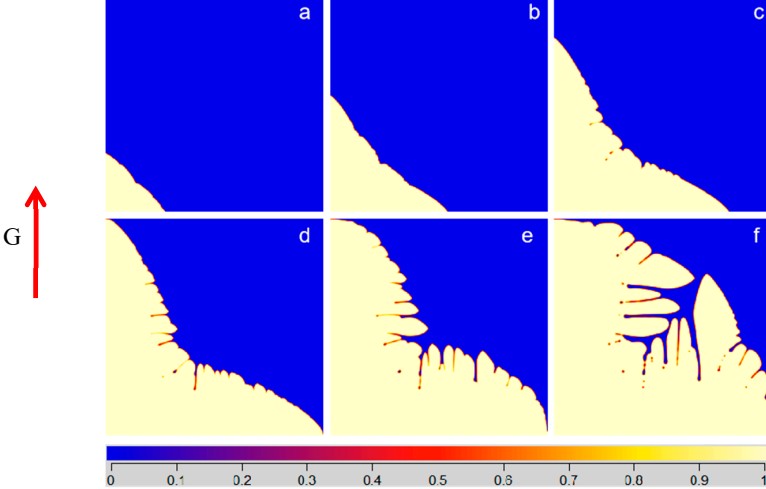

**Figure 3.** Grain growth with time of Ti–3Al–0.9Fe alloy: (**a**) 4 ms, (**b**) 8 ms, (**c**) 12 ms, (**d**) 16 ms, (**e**) 20 ms, (**f**) 30 ms.

The influence of the increase in Fe content on the component supercooling was discussed. The existence of supercooling zone depends on the temperature gradient at the solid–liquid interface determined by the external heat flux,

$$G = \frac{dT_L}{dx'}\bigg|_{x'=0} \tag{4}$$

where $G$ is the temperature gradient at the solid–liquid interface determined by the external heat flux; $T_L$ is the actual temperature of the liquid phase at the front of the interface; and $x'$ is the direction of the temperature gradient. In equilibrium, there is $G = -mG_c$, where $G_c$ is the concentration gradient. When $G \geq G_c$, the liquid phase in the front interface is in the state of component supercooling. According to the study of Kurz et al. [23], by assuming that there is no convection in the liquid phase and only diffusion, the critical condition of component supercooling can be rewritten as

$$\frac{G}{v} \geq \frac{mC_0(1-k_0)}{Dk_0} \tag{5}$$

where $m$ is the slope of liquidus; $D$ is the diffusion coefficient of liquid phase; $v$ is the migration rate of interface; $C_0$ is the initial composition; and $k_0$ is the distribution coefficient. For the Ti–6Al–4V–xFe alloy in this paper, if Ti is the solvent and Al, V, and Fe are the solute, then the liquid surface is a function of the concentration of Al, V, and Fe in the liquid phase, $T_L = T_L(C_{Al}, C_V, C_{Fe})$. The temperature gradient of the liquid melting point at the solid–liquid interface is:

$$\frac{dT_l}{dx'}\bigg|_{x'=0} = m_{Al}\left(\frac{dC_{Al}}{dx'}\right)_{x'=0} + m_V\left(\frac{dC_V}{dx'}\right)_{x'=0} + m_{Fe}\left(\frac{dC_{Fe}}{dx'}\right)_{x'=0} \tag{6}$$

where $m_{Al}$ is slope of the liquidus of $C_{Al}$, $m_{Al} = \frac{dT_L}{dC_{Al}}$, $m_V = \frac{dT_L}{dC_V}$, and $m_{Fe} = \frac{dT_L}{dC_{Fe}}$.

In the equilibrium state, the solute mass at the solid–liquid interface is conserved. Assuming that there is no interaction between Al, V, and Fe, there is

$$D_{Al}\left(\frac{dC_{Al}}{dx'}\right) = -v\left(\frac{C_{0Al}}{k_{Al}} - C_{0Al}\right) \tag{7}$$

$$D_V\left(\frac{dC_V}{dx'}\right) = -v\left(\frac{C_{0V}}{k_V} - C_{0V}\right) \tag{8}$$

$$D_{Fe}\left(\frac{dC_{Fe}}{dx'}\right) = -v\left(\frac{C_{0Fe}}{k_{Fe}} - C_{0Fe}\right) \tag{9}$$

where $D_{Al}$, $D_V$, and $D_{Fe}$ are the liquid diffusion coefficients of the corresponding element; $C_{0Al}$, $C_{0V}$, and $C_{0Fe}$ are the initial concentrations of the corresponding element; $k_{Al}$, $k_V$, and $k_{Fe}$ are the partition coefficients of the corresponding elements. Substitute Equations (7)–(9) into Equation (6), and the actual temperature gradient G is greater than or equal to $\frac{dT_l}{dx'}\big|_{x'=0}$:

$$\frac{G}{v} \geq -\frac{m_{Al}C_{0Al}(1-k_{Al})}{D_{Al}k_{Al}} - \frac{m_V C_{0V}(1-k_V)}{D_V k_V} - \frac{m_{Fe}C_{0Fe}(1-k_{Fe})}{D_{Fe}k_{Fe}} \tag{10}$$

According to Equation (10), due to $k_{Fe} < 1$, the component supercooling is easier to achieve when $C_{0Fe}$ increases. Therefore, the increase in Fe content will promote the formation of the component supercooling zone, which will affect the morphology of the grains.

### 3.2. Effect of Fe Content on the Microstructure of Multiple Grains

The growth of several grains with different Fe content was simulated by MICRESS. Figure 4 shows the effect of Fe content on grain size (the color bar represents the mass fraction of Al). With higher Fe content, the shape of grains is more complex and the grain size is more refined. Due to the low

directional temperature gradient in the levitation melting, the overall appearance of equiaxed crystal appears. The crystal interface is always composed of crystal faces with smaller interface energy. The interface energy is smaller at the wide crystal face, while the energy of narrow crystal face at the edge is larger. Therefore, the crystal morphology tends to be spherical polyhedron in a stable state. Figure 5 shows as the time goes on, the liquid phase almost disappeared at 0.85 s, and an equiaxed crystal with larger grains was obtained. For the titanium alloy, the BCC phase of the cubic crystal system was first formed during solidification, and the optimal growth direction was the <001> crystal direction.

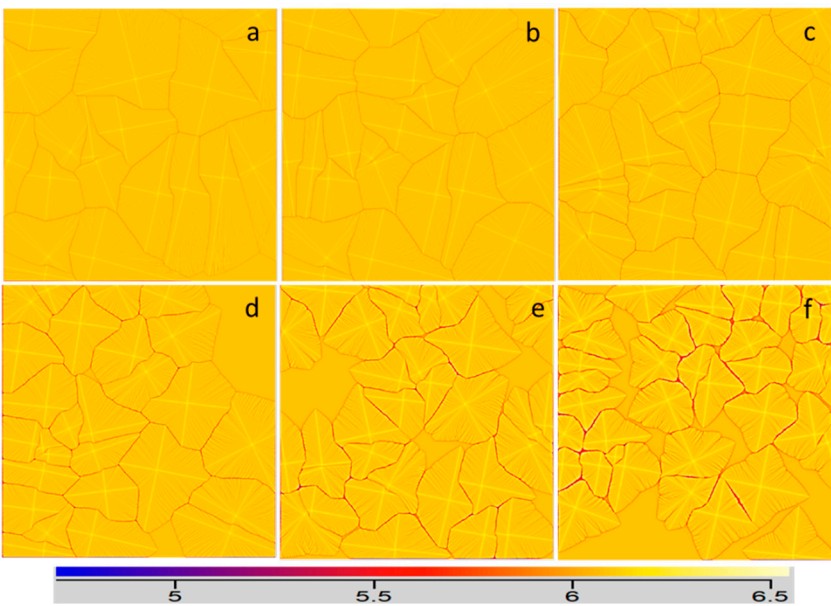

**Figure 4.** Effect of Fe content on grain size at 2 s: (**a**) Ti–6Al–4V–0Fe, (**b**) Ti–6Al–4V–0.1Fe, (**c**) Ti–6Al–4V–0.3Fe, (**d**) Ti–6Al–4V–0.5Fe, (**e**) Ti–6Al–4V–0.7Fe, (**f**) Ti–6Al–4V–0.9Fe.

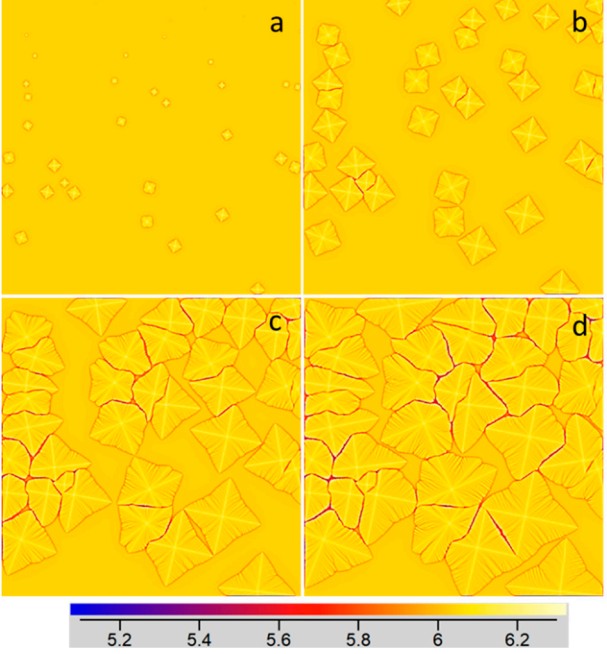

**Figure 5.** The grain growth with mass fraction of Fe is 0.9: (**a**) 0.3 s, (**b**) 0.8 s, (**c**) 1.5 s, (**d**) 2.0 s.

For the Ti–6Al–4V–xFe alloy, there was a large solute concentration gradient in the solid–liquid interface at the front edge of the polyhedron, and its diffusion rate was faster than that of the large plane crystal surface with a small solute concentration gradient at the front edge of the interface, resulting in the gradual change of the crystal from an octahedron to a star. This trend was more obvious at the region with a higher Fe content. Compared with Figure 4d, the segregation of Fe at the front of the solid–liquid interface in Figure 4f was stronger, and the resulting local supercooling slowed down the interface migration rate.

The microstructure of each direction was very different due to the different influence of the solute diffusion field and temperature diffusion field in four <001> directions. Figure 6 shows the effect of Fe content on the grain growth rate. When the Fe content exceeds 0.3 wt%, the growth rate of the alloy begins to decrease significantly. If the Fe content reaches 0.9 wt%, more time is needed for the liquid phase to disappear.

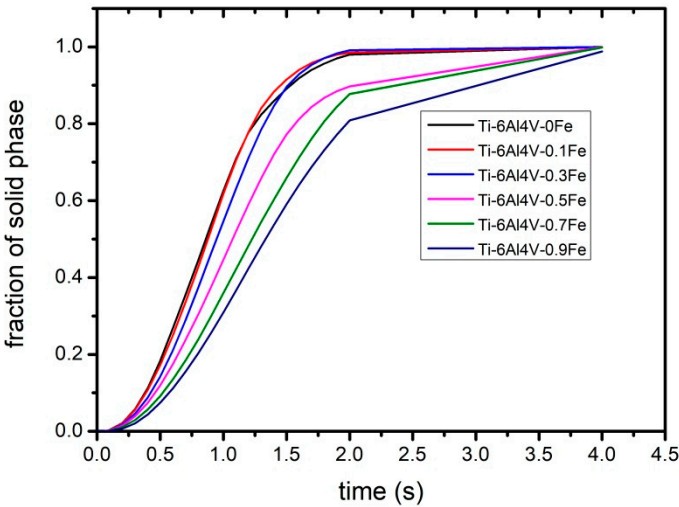

**Figure 6.** Effect of Fe content on grain growth rate.

In the growth process, the gap between the grains is large, and the growth speed of the grains is slow, which may provide more space for the growth of small grains and reduce the annexation of grains. Therefore, the increase of Fe in the experiment made the grains more refined.

Figure 7 shows the grain size of the alloy obtained by levitation melting. In a certain range, with the increase in Fe content, the grain size of the alloy gradually decreases. According to the number of grains and the cut-off area, the average grain radius is simply estimated, as shown in Figure 8. Compared with the simulated grain size, the experimental result was larger, which is due to the limited simulation time, while the experimental grains completed the grain growth. When there was no Fe in the alloy, as shown in Figure 7a, the grain size was the largest and the grain distribution was relatively uniform. The grain radius was about 2.29 mm, and the shape of the grain was close to circular. With the increase in Fe content, the grain size of the alloy decreased gradually, while the overall decreasing trend was mitigated. When the Fe content was 0.9 wt%, the average grain radius was the smallest (about 1.03 mm).

With the increase in Fe content, the distribution of grains was no longer uniform. Some small grains were distributed at the junction of larger grains, and the morphology of grains was close to a complex polygon. It can be considered that the addition of Fe changes the size and distribution of the grains and affects the shape of the grains, which verifies the simulation results.

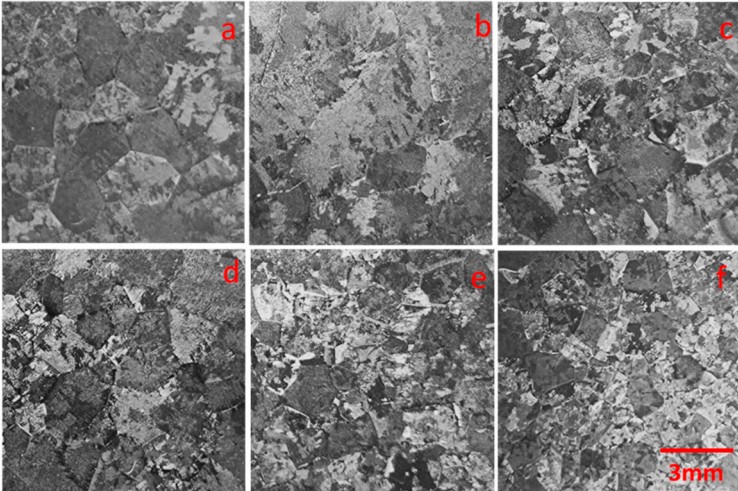

**Figure 7.** Effect of Fe content on grain size: (**a**) 0Fe, (**b**) 0.1Fe, (**c**) 0.3Fe, (**d**) 0.5Fe, (**e**) 0.7Fe, (**f**) 0.9Fe.

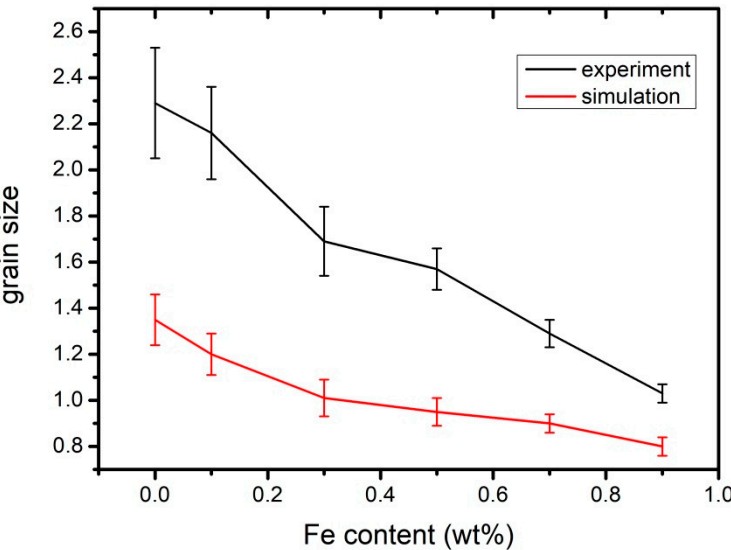

**Figure 8.** The change in the average grain size with Fe content.

### 3.3. Element Distribution in Ti–6Al–4V–xFe Alloy

As shown in Figure 9, the Fe composition distribution along the green lines was obtained and demonstrated in Figure 9e. As the Fe content increased from Figure 9a–d, the maximum solute concentration $C_L{}^*$ of Fe in the liquid phase at the solid–liquid boundary continued to rise (here represented by mass fraction), which were 0.67, 1.12, 1.48, and 1.63 wt%, respectively, corresponding to the four peaks in Figure 9e. The segregation ratio $S_R$ was 4.01, 4.15, 4.00, and 3.54, respectively, and the overall segregation trend was reduced. Within a certain range, the diffusion distance $\delta_n$ of Fe (Figure 9f) in the liquid phase had a linear relationship with the Fe content in the alloy, and the relationship can be fitted as:

$$\delta_n = 31.1C_0 + 29.5 \qquad (11)$$

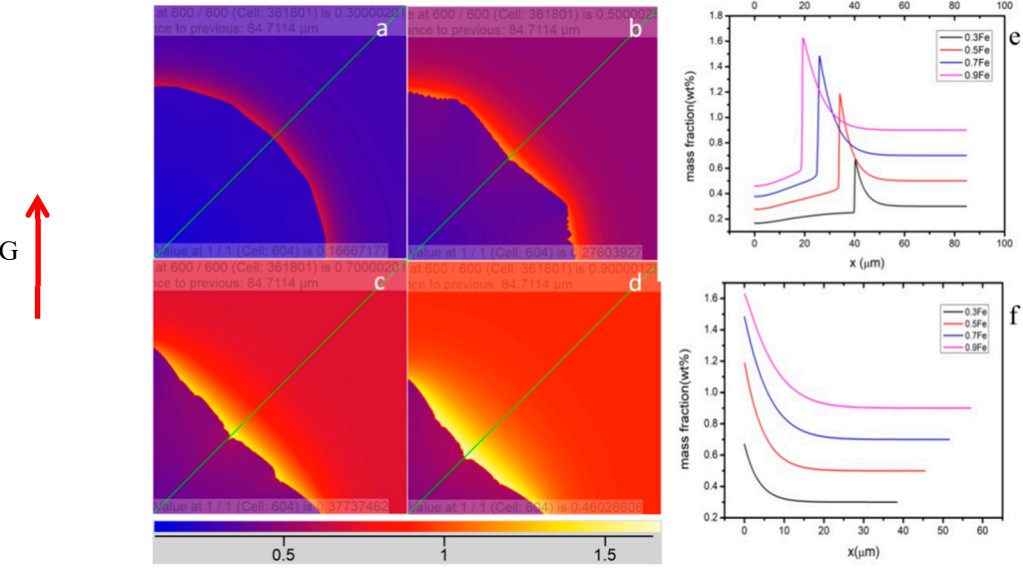

**Figure 9.** The change of Fe mass fraction in the direction perpendicular to the larger plane of grain under different Fe content: (**a**) 0.3Fe, (**b**) 0.5Fe, (**c**) 0.7Fe, (**d**) 0.9Fe, (**e**) liquid and solid phase, (**f**) liquid phase.

According to classical theory, for convective solute distribution, under directional solidification conditions, there is:

$$D_L \frac{d^2 C_L}{dx^2} + v \frac{d_{C_L}}{dx'} = 0 \tag{12}$$

when $x = 0$, $C_L = C_L^* < C_0/k_0$, and when $x = \delta_n$, $C_L = C_0$.

Defining $\frac{dC_L}{dx} = z$, $\frac{dz}{dx} = \frac{d^2 C_L}{dx^2}$, then $\frac{dz}{z} = -\frac{v}{D_L} d$. After inserting the boundary conditions into the function, we can obtain:

$$C_L = \left(1 - \frac{1 - e^{-\frac{v}{D_L} x}}{1 - e^{-\frac{v}{D_L} \delta n}}\right)(C_L^* - C_0) + C_0 \tag{13}$$

where $k_0$ is the partition coefficient; $x$ is the diffusion distance; $v$ is the interface moving rate; and $D_L$ is the liquid diffusion coefficient.

Three assumptions were made: (1) there is only diffusion (no convection) in the liquid phase; (2) the diffusion distance $\delta_n$ at the thin solid–liquid interface are infinity; and (3) the components of the liquid phase outside the solute enrichment layer keep the original concentration $C_0$ unchanged during the solidification process. Under these assumptions, the maximum solute concentration is $C_L^* = C_0/K_0$ in the stable liquid phase, and the solute distribution equation of the stable state can be simplified as:

$$C_L = C_0 \left[1 + \left(\frac{1 - k_0}{k_0}\right) e^{-\frac{v}{D_L} x}\right] \tag{14}$$

The composition of the liquid phase outside the solute enrichment layer is no longer $C_0$, but gradually increases in the case of limited liquid volume due to convection in the outer diffusion layer during the actual solute redistribution process. As the actual $C_L^* < C_0/K_0$, the solute concentration calculated by Equation (13) is higher, as shown in Figure 10. In this work, since the solidification process of levitation melting was not directional solidification, the direction of the temperature gradient has little effect on the grain morphology. As shown in Figure 9, the grain growth speed was slow in the direction perpendicular to the larger plane of the grain, which had an angle of 45° with the relative temperature gradient direction. The solute distribution value should be between Equations (13) and (14). For the levitation melting of the Ti–6Al–4V–xFe alloy with a slow growth rate, the modified

distribution equation of solute in the steady state can be proposed according to the results of phase field simulation:

$$C_L = 0.79C_0 \left[ 1 + \left( \frac{1-k_0}{k_0} \right) e^{-\frac{v}{D_L}x} \right] + 0.11 \tag{15}$$

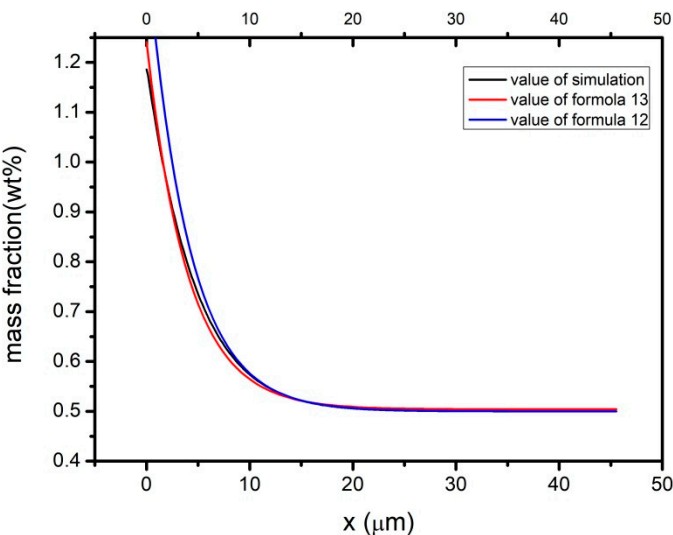

**Figure 10.** Solute distribution of Ti–6Al–4V–0.5Fe in the liquid phase.

The agreement of Equation (14) with the simulation results was close to 90%, while the agreement of the modified equation with the simulation results was close to 97%.

The composition at the triangular grain boundaries of Ti–6Al–4V–0.5Fe and Ti–6Al–4V–0.9Fe alloys were scanned by EPMA, and the results are shown in Figure 11. The segregation of Ti at the grain boundary was not obvious. The overall distribution presents a homogeneous contrast due to the matrix material of Ti. Compared with Figure 11a, Figure 11b shows that there was a certain segregation of the Al element in $\alpha$ lamellae. The most serious segregation was in the $\beta$ grain boundary, while the lowest content was at the edges of the $\beta$ grain boundary.

In the solidification process of the titanium alloy, the solid–liquid phase transformation first occurs, forming $\beta$ original grains, and growing continuously with the decrease in temperature. The amount of liquid phase gradually decreases and concentrates at the boundary of $\beta$ grains at the end of the solid–liquid phase transformation, as presented in Figure 5d. As the solidification proceeded, the liquid phase finally disappeared, forming the original $\beta$ grain, as indicated in Figure 4a. With the slow decrease in temperature (i.e., non-quenching), the BCC phase in the high temperature state of the Ti alloy was gradually transformed into the HCP phase (i.e., $\beta/\alpha$ transformation, forming primary $\alpha$ phase). The $\alpha$ lamellar structure (about 0.5–2 $\mu$m) was formed in the original $\beta$ grain; and the remaining $\beta$ phase was distributed at the boundary of the $\alpha$ lamellar. The morphology of the $\beta$ original grain was retained without any deformation in the end. As a result, the microstructure shown in Figure 11a was formed. During the cooling process, a relatively wide $\alpha$ lamellar structure (about 2–3 $\mu$m) was formed from the $\beta$ grain boundaries. The remaining $\beta$ phase was distributed at the edges.

In the same way, V and Fe, as $\beta$ stable elements, concentration increased from the inner area to the edges of $\alpha$ lamellae. Due to the wide $\beta$ grain boundary, the segregation at the edge of the $\beta$ grain boundary was more obvious. Since Fe is a stronger $\beta$ stable element than V, the segregation of Fe was more obvious. Comparing the $\beta$ grain boundaries in Figure 11a,c, Figure 11c was finer (about 1–2 $\mu$m), which may be attributed to the grain refinement of Fe. The distribution trend of Figure 11b was the same as in Figure 11d.

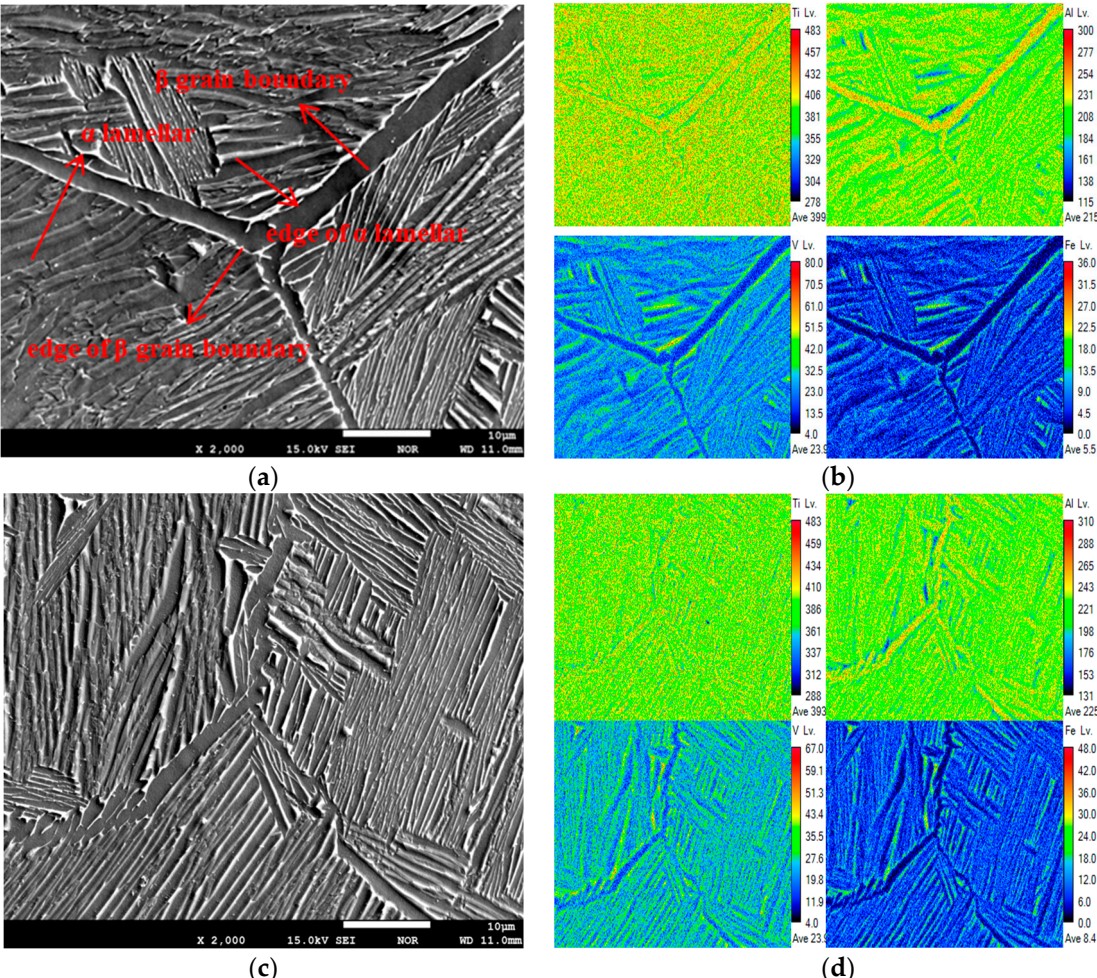

**Figure 11.** Results of the electron probe micro analysis surface scan. (**a**) Secondary electron image of Ti–6Al–4V–0.5Fe, (**b**) Concentration distribution of Ti–6Al–4V–0.5fe, (**c**) Secondary electron image of Ti–6Al–4V–0.9Fe, (**d**) Concentration distribution of Ti–6Al–4V–0.9Fe.

Figure 12 compares the simulated with the experimental values of the Fe composition distribution. As the simulation process does not complete the β/α transformation, the segregation of Fe is mainly concentrated in the residual liquid phase between β grains. Comparing the segregation degree of the simulated and experimental values, the segregation degree of Fe in the simulation was not more than three times that of the nominal composition, whereas the segregation degree of the experimental value was close to six times the nominal composition. This may be attributed to the decrease in β phase amount in β/α transformation and the further compression of the range of Fe segregation distribution.

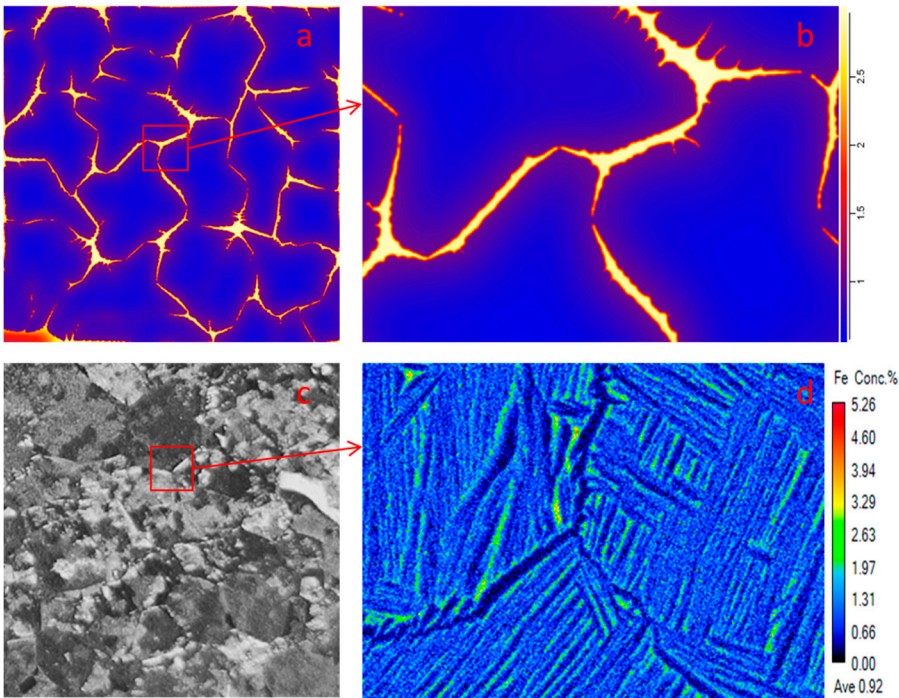

**Figure 12.** Comparison of simulated and experimental values of Fe concentration in Ti–6Al–4V–0.9Fe alloy. (**a**) Selection of simulation position, (**b**) Simulated value of Fe concentration, (**c**) Selection of experimental position, (**d**) Experimental value of Fe concentration.

Through line scans crossing the grain boundaries of Ti–6Al–4V–0.5Fe, Ti–6Al–4V–0.7Fe, and Ti–6Al–4V–0.9Fe alloys, it can be seen from Figure 13b that the fluctuation range of Al composition in the alloy ranged from 9.78 to 11.07 at%, V ranged from 2.19 to 7.98 at%, and Fe from 0.18 to 1.81 at%. Compared with the average composition, the fluctuation values of Al, V and Fe were 8%, 91%, and 202%, respectively. Obviously, the segregation of Fe was greater than V, while the segregation of V was greater than Al.

For three samples, the mean values of Fe composition were 0.60 at%, 0.76 at%, and 0.87 at% with the standard deviations of 0.31, 0.39, and 0.40, respectively. Considering that the grain boundary of the Ti–6Al–4V–0.9Fe sample was less and the overall element distribution was more uniform, the segregation of Fe was still slightly larger. It can be seen that in a certain range, with the increase in Fe content in the alloy, the segregation of Fe tended to increase; however, the influence of Fe content on the segregation of Al and V elements was negligible. With the increase in Fe content, the trends of Fe segregation in the simulation and experiment were the opposite. It is considered that the segregation of Fe mainly occurs in the stage of grain growth or solid-state phase transformation, which needs further study.

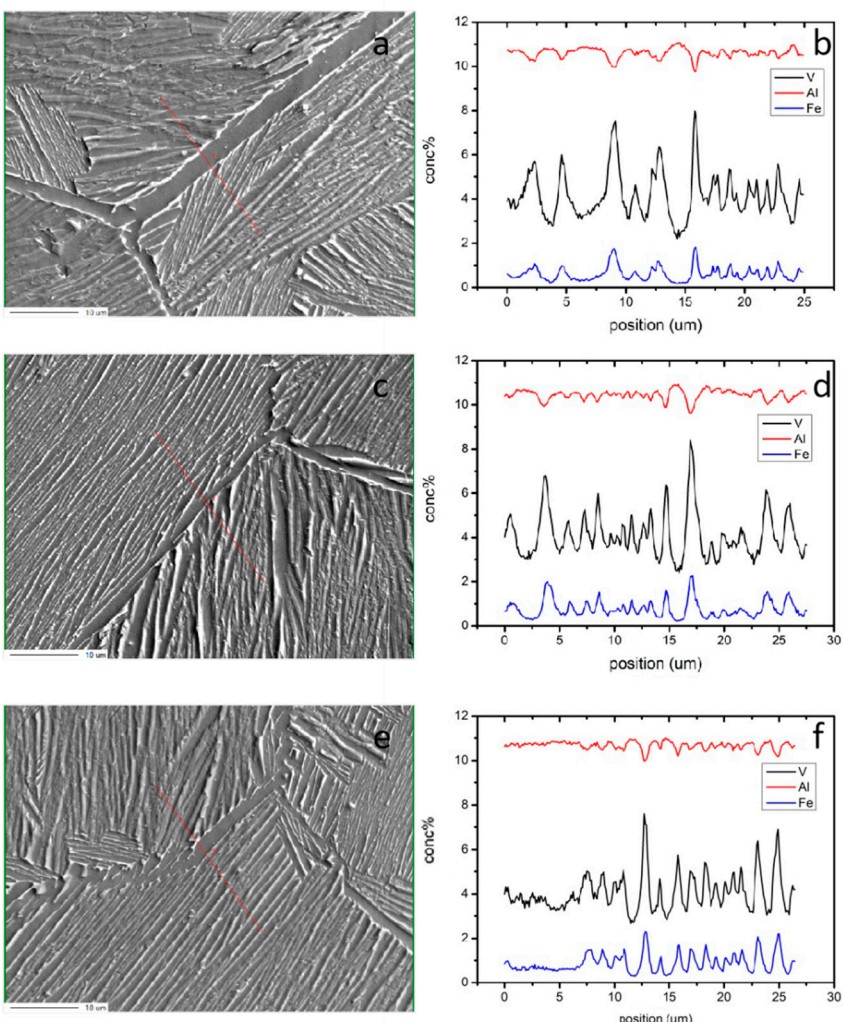

**Figure 13.** Results of the EPMA line scan. (**a**) Selected location of Ti–6Al–4V–0.5Fe, (**b**) Composition distribution of Ti–6Al–4V–0.5Fe, (**c**) Selected location of Ti–6Al–4V–0.7Fe, (**d**) Composition distribution of Ti–6Al–4V–0.7Fe, (**e**) Selected location of Ti–6Al–4V–0.9Fe, (**f**) Composition distribution of Ti–6Al–4V–0.9Fe.

## 4. Conclusions

In this work, the processes of levitation melting of five Ti–6Al–4V–xFe alloys were simulated, the effect of Fe content on the microstructure of single crystal and multi crystal was studied, and the distribution of elements in the Ti–6Al–4V–xFe alloy was discussed. Some simulation results were verified by experiments. The specific conclusions are as follows:

(1)  The segregation of Fe element at the grain boundary of Ti–6Al–4V–xFe alloys can inhibit the interface mobility, thus promoting the formation of a local supercooling zone and making the grains easier to grow into dendrites.

(2)  With the increase of Fe content, the grain size of the alloy decreased gradually. When there was no Fe in the alloy, the grain size was the largest (radius close to 2.29 mm), the grains were more uniform, and the shape of the grain was close to circular. The grain size decreased gradually with an increase in the Fe content and the overall decrease trend slowed down. When the Fe content was 0.9, the average grain radius was the smallest, which was about 1.03 mm.

(3)  With the increase in Fe content, the distance of diffusion layer $\delta_n$ increased in the liquid phase. Within a certain range, there was a linear relationship between them.

(4)  The segregation of Fe was more obvious than that of Al and V. With the increase in Fe content, the segregation of Fe increased, but there was less of an effect on Al and V.

**Author Contributions:** Conceptualization, H.C.; data curation, Y.G.; formal analysis, L.D.; funding acquisition, L.Z.; investigation, D.Z.; methodology, F.C.; software, L.D.; supervision, R.H. All authors have read and agreed to the published version of the manuscript.

**Funding:** This work was supported by Primary Research and Development Plan of Jiangsu Province (BE2019119).

**Conflicts of Interest:** The authors declare no conflict of interest.

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
