# Peer review of "Effect of Fe Content on the As-Cast Microstructures of Ti–6Al–4V–xFe Alloys"

_metals, doi:10.3390/met10080989_

Round 1

Reviewer 1 Report

Modeling description should be significantly improved to make the reader better understand. First, a schematic representation of the modelled domain should be reported. In addition, all the adopted equations along with boundary conditions, parameters and references should be also reported. Information (e.g. vendor) about the micress 6.3 software should be added.

Author Response

Response to Reviewer 1 Comments

Modeling description should be significantly improved to make the reader better understand. First, a schematic representation of the modelled domain should be reported. In addition, all the adopted equations along with boundary conditions, parameters and references should be also reported. Information (e.g. vendor) about the micress 6.3 software should be added.

Response:

Thank you for your comments. I've added a schematic about the modeled domain.

I have modified the relevant model and added the diffusion equation, and some explanations of the model are added as shown in page 2 and 3. According to your request, I added the information about micress.

Reviewer 2 Report

A very good paper, but:

  1. Why did the authors decided to choose to complete Ti6Al4V with Fe, knewing that V and Al are hardly avoided by scientific/medical world, because of secondary effects (Alzheimer disease) and because of low biocomptability explain in a lot of scientific papers????
  2. What were the basis to choose Fe instead of other MORE biocompatible chemical elements such Sn, Nb, Zr, Ta and so on…. ????
  3. Why the authors added so little amount of Fe … below 1% …???!!! Which are the arguments to do that instead to study an amount of Fe up to 1 or 2 or 5 %?

The research is not useful for medical world taking into consideration the effects of Al and V over the human body. Good work, but useless… please reject!

Author Response

Response to reviewer2

A very good paper, but:

Why did the authors decided to choose to complete Ti6Al4V with Fe, knewing that V and Al are hardly avoided by scientific/medical world, because of secondary effects (Alzheimer disease) and because of low biocomptability explain in a lot of scientific papers????

1 、What were the basis to choose Fe instead of other MORE biocompatible chemical elements such Sn, Nb, Zr, Ta and so on…. ????

Response1:Thank you for your comments,I'm sorry that I may not be able to give you any instructions. The titanium alloy I studied is mainly used as a structural material in the aerospace field rather than in the biological field. Iron is selected because the addition of Fe can effectively refine the grain size and improve the mechanical properties of the alloy, which has been mentioned in the introduction.

2、Why the authors added so little amount of Fe … below 1% …???!!! Which are the arguments to do that instead to study an amount of Fe up to 1 or 2 or 5 %?

Response2:It is introduced in the introduction that adding Fe to titanium alloy may cause beta spot, which will have side effect on the plasticity of titanium alloy. According to the research of our group, adding no more than 1% Fe into Ti-6Al-4V can not only improve the strength, but also have no great influence on the plasticity. My research is also based on this.

3、The research is not useful for medical world taking into consideration the effects of Al and V over the human body. Good work, but useless… please reject!

Response3:As stated in response 1, the alloy is not used in biological field, so it is not necessary to consider the effect on human body.

Reviewer 3 Report

Reviewer’s comments:

In this manuscript, the authors conducted phase-field simulation and experimental characterization to investigate the effects of Fe on the solidification of Ti-6Al-4V alloys. Overall the manuscript is very difficult to digest and follow although some of the results could be potentially valuable and useful. The contents of the manuscript are poorly organized and polished. Many figures are quite vague and do not show clear evidence to support the arguments made in the manuscript. Therefore, I would suggest rejection of the manuscript in its current form.

  • Page 2. The sentence “As the temperature change….” appeared twice, where the second appearance is clearly inappropriate and should be a careless mistake due to editing.
  • Page 2, the description of Eq. 1 is unclear. Please clarify Delta_G represents which phase’s Gibbs energy. Or it represents the Gibbs energy difference between which two phases? The author should also give the value of the order parameter for each phase.
  • Page 4, Table 2, there are Chinese characters in the table, which is very inappropriate and unprofessional.
  • The authors claimed that the temperature gradient in their simulations is small but never gave a specific value.
  • The color bars in Figs. 1-5 should be labeled.
  • Page 5. What are the purposes of listing Eq. 2-8? It is very confusing the authors listed those equations but never used them for calculation or discussion.
  • Page 6, please clarify what is the meaning of “Fe concentration is stable”
  • Page 6, Please provide more details to explain why the results of Fig. 3 support the argument “whereas when the Fe concentration changes greatly, the interface mobility changes accordingly”
  • Page 7, the authors claimed that the solid phase should be isotropic but without providing any results or citations to support the argument.
  • Color scales of the elemental distributions in Fig. 11 should be provided.
  • Based on the results of Fig. 11b and 11d, Fe seems to prefer to segregate at the interface between the alpha and beta phases instead of the grain boundaries.
  • Page 15, Fig. 13 is mistakenly listed as Fig. 7-37.

Author Response

Response to reviewer3

In this manuscript, the authors conducted phase-field simulation and experimental characterization to investigate the effects of Fe on the solidification of Ti-6Al-4V alloys. Overall the manuscript is very difficult to digest and follow although some of the results could be potentially valuable and useful. The contents of the manuscript are poorly organized and polished. Many figures are quite vague and do not show clear evidence to support the arguments made in the manuscript. Therefore, I would suggest rejection of the manuscript in its current form.

Page 2. The sentence “As the temperature change….” appeared twice, where the second appearance is clearly inappropriate and should be a careless mistake due to editing.

Response1:Thank you for your comments, I have made relevant changes

Page 2, the description of Eq. 1 is unclear. Please clarify Delta_G represents which phase’s Gibbs energy. Or it represents the Gibbs energy difference between which two phases? The author should also give the value of the order parameter for each phase.

Response2:Thank you for your advice, I have adjusted the whole model at page 3

Page 4, Table 2, there are Chinese characters in the table, which is very inappropriate and unprofessional.

Response3:I’m sorry for this mistake, and I have corrected it.

The authors claimed that the temperature gradient in their simulations is small but never gave a specific value.

Response4:I have added the temperature gradient at page 5.

The color bars in Figs. 1-5 should be labeled.

Response5:I have explained the color bar at page 5 and page 7.

Page 5. What are the purposes of listing Eq. 2-8? It is very confusing the authors listed those equations but never used them for calculation or discussion.

Response6:Thank you for your suggestion, I have added the relevant discussion at page 7.

Page 6, please clarify what is the meaning of “Fe concentration is stable”

Page 6, Please provide more details to explain why the results of Fig. 3 support the argument “whereas when the Fe concentration changes greatly, the interface mobility changes accordingly”

Response7:Indeed, this graph does not effectively explain the correlation between mobility and Fe mass fraction. After discussion with the co authors, I decided to delete this part.

Page 7, the authors claimed that the solid phase should be isotropic but without providing any results or citations to support the argument.

Response8:As you said, I decided to delete this argument.

Color scales of the elemental distributions in Fig. 11 should be provided.

Response9:As you suggested, I've added the color bar.

Based on the results of Fig. 11b and 11d, Fe seems to prefer to segregate at the interface between the alpha and beta phases instead of the grain boundaries.

Response10:Indeed, I have pointed out in the paper that the original beta grain boundary has actually become a larger α-lamella due to the solid-solid phase transformation of β phase during cooling, so Fe is distributed at the edge of β grain boundary.

Page 15, Fig. 13 is mistakenly listed as Fig. 7-37.

Response11:I have revised the title of the picture according to your request

Round 2

Reviewer 1 Report

The revised version of the manuscript can be accepted for publication.

Reviewer 2 Report

The answers clarifies issues on first round. It can be accepted with all corrections made.